# Exposure of Cultured Hippocampal Neurons to the Mitochondrial Uncoupler Carbonyl Cyanide Chlorophenylhydrazone Induces a Rapid Growth of Dendritic Processes

**DOI:** 10.3390/ijms241612940

**Published:** 2023-08-18

**Authors:** Liliia Kushnireva, Eduard Korkotian, Menahem Segal

**Affiliations:** 1Faculty of Biology, Perm State University, 614068 Perm, Russia; lilikushnireva@psu.ru; 2Department of Brain Sciences, Weizmann Institute of Science, Rehovot 7610001, Israel; eduard.korkotian@weizmann.ac.il

**Keywords:** carbonyl cyanide chlorophenylhydrazone (CCCP), calcium, Orai1, STIM1, dendritic processes, mitochondria

## Abstract

A major route for the influx of calcium ions into neurons uses the STIM-Orai1 voltage-independent channel. Once cytosolic calcium ([Ca^2+^]i) elevates, it activates mitochondrial and endoplasmic calcium stores to affect downstream molecular pathways. In the present study, we employed a novel drug, carbonyl cyanide chlorophenylhydrazone (CCCP), a mitochondrial uncoupler, to explore the role of mitochondria in cultured neuronal morphology. CCCP caused a sustained elevation of [Ca^2+^]i and, quite surprisingly, a massive increase in the density of dendritic filopodia and spines in the affected neurons. This morphological change can be prevented in cultures exposed to a calcium-free medium, Orai1 antagonist 2APB, or cells transfected with a mutant Orai1 plasmid. It is suggested that CCCP activates mitochondria through the influx of calcium to cause rapid growth of dendritic processes.

## 1. Introduction

Variations in cytosolic calcium concentrations ([Ca^2+^]i) play a crucial role in neuronal growth, communication, plasticity and survival. [Ca^2+^]i is accumulated in calcium stores, including endoplasmic reticulum and mitochondria, so as to maintain a very low ambient [Ca^2+^]i (4–5 orders of magnitude below extracellular [Ca^2+^]). This is controlled with voltage and ligand-gated calcium entry and with extrusion channels. Recent evidence assigned a pivotal role of stromal interaction molecule 1 (STIM1) in the regulation of [Ca^2+^]i [1]. Accordingly, STIM1 clusters near the depleted store and relocates to the membrane, where it activates the Orai1 plasma membrane voltage-independent calcium channel to allow calcium influx into the cell to refill the stores [2]. The interaction of stores/STIM/Orai has been studied extensively in non-neuronal cells [3]. Relatively less is known about their role in central neurons. STIM1 and Orai1 are localized in the brain [4] and can be converted from a dispersed to a punctate form upon depletion of calcium stores with thapsigargin [5]. They are important in the regulation of growth cone motility [6,7], in the regulation of voltage-gated calcium channels [8] and in the detrimental effects of chronic epilepsy [9], ischemia [10] and oxidative stress [11]. While STIM1 has been imaged in a punctate form in the dendrites of hippocampal neurons [12], the dynamic distribution and function of Orai1 in such neurons remained enigmatic [13]. In the present study, we combined time-lapse imaging with pharmacological tools to explore the role of Orai1 in cultured hippocampal neurons. We used a mitochondria uncoupler carbonyl cyanide chlorophenylhydrazone (CCCP), which increases proton permeability and depolarizes mitochondrial membrane potential. Previously, we found that CCCP causes an increase in the frequency of spontaneous global calcium bursts with their subsequent complete disappearance [14]. We now demonstrate that longer exposure to CCCP induces [Ca^2+^]i rise in an Orai1-dependent manner, conjugates mitochondrial calcium saturation and massive growth of dendritic processes, followed by apoptotic swelling and subsequent death of the cultured hippocampal neurons.

## 2. Results

In the first series of experiments, we explored the long-term effects of CCCP on [Ca^2+^]i and the formation of novel dendritic filopodia/spines. As can be seen in Figure 1, [Ca^2+^]i, measured with the membrane permeant Fluo-2 AM, grows gradually over 30 min of exposure (Figure 1A–C). In the same group of neurons, a significant increase in the density of dendritic processes is clear at 30 min of exposure to the drug (Figure 1A,D). To obtain a direct indication of the role of extracellular calcium ([Ca^2+^]o) to change in [Ca^2+^]i, the cells were maintained in nominally 0 mM calcium in the imaging medium. In these conditions, there was no change in [Ca^2+^]i and no change in filopodia/spines density (Figure 1E–G). Upon replenishment of 2 mM calcium in the medium, there was a rapid increase in [Ca^2+^]i and a correlative increase in filopodia/spines density (Figure 1E–G).

Several possible routes for the entry of calcium into the neurons may facilitate the formation of dendritic filopodia. To identify the route of calcium entry under the CCCP conditions, we exposed the cultures to different calcium antagonists, including APV, TTX and DNQX, nifedipine and conotoxin, as well as 2-APB, an antagonist of the Orai1-2 channels. The CCCP-induced increase in [Ca^2+^]i and growth of dendritic processes was completely blocked (average 17.26 ± 1.43 processes in control conditions and 17.65 ± 1.48 after 40 min with CCCP + 2-APB per 50 µm dendritic segments, not significant, *t*-test, *n* = 23 cells) by the use of 2-APB (Figure 2A,B). Furthermore, cells transfected with the mutant Orai1(mOrai1) were resistant to the CCCP-induced rise in [Ca^2+^]i (Figure 2E–G). In the presence of mOrai1, CCCP led to only a slight increase in cytosolic calcium (Figure 2G) and did not lead to a significant increase in processes (Figure 2E,F). This indicates that CCCP is using the Orai channel to load the cell with [Ca^2+^]. It should be noted that although other antagonists did not affect the overall increase in cytosolic calcium and the growth of new processes, some of them had their own isolated effects. Thus, the application of ω-conotoxin and nifedipine led to a short-term burst of network activity, probably due to the inhibition of GABAergic neurons [15]. This led to a temporary increase in [Ca^2+^]i and a subsequent return to the baseline. In contrast, the use of thapsigargin caused a rapid but sustained increase in intracellular Ca^2+^, which was not limited to the drug preincubation period. As a result, the action of CCCP began not from the basal but from an already elevated [Ca^2+^]i, which created the illusion of a suppression sign of the effect of thapsigargin (Figure 2C). In fact, the total growth of [Ca^2+^]i in thapsigargin + CCCP was the same as in CCCP alone, and the outgrowth of protrusions was not reduced to a statistically significant extent.

As a result of a small increase in cytosolic calcium that occurs in the experiment with mOrai1 (Figure 2G), there was a slight tendency for the growth of new processes; however, this increase was not sufficient for a significant difference; mOrai1/mOrai1 + CCCP 30 min: 15.87 ± 0.80/16.22 ± 0.86 processes, *n* = 16 cells, *t*-test. Since after a strong increase in the cytosolic calcium level, the number of processes significantly increased, as shown in Figure 1A,D,F(bottom),G, it can be concluded that dendritic processes grow due to a sufficient increase in cytosolic calcium levels following CCCP-treatment. Interestingly, this effect did not appear when CCCP was extensively washed out after 5 min of incubation (Figure 2D): control/CCCP wash: 15.23 ± 1.01/13.69 ± 0.84, not significant, *n* = 16 cells, *t*-test. By itself, early washout did not prevent further neurotoxicity. New protrusion outgrowth and, finally, the appearance of dendritic blubs, the first signs of neurotoxicity and cell death, are shown in Appendix A. Thus, the appearance of new processes under the influence of CCCP is not directly related to its toxic effect on cells.

To further explore CCCP effects through the STIM-Orai pathway, we transfected neurons with normal Orai1, and in these neurons, the presence of CCCP also led to the growth of new processes (Figure 3A,B), similar to the previous experiment in control conditions (Figure 1D). We also showed that the process growth with CCCP does not depend on the characteristics of eBFP as a morphological marker since an experiment was carried out with another morphological marker, DsRed (Figure 3A). This indicates that the transfection itself is not responsible for the increase in [Ca^2+^]i or the density of newly-formed protrusions. We also observed an insignificant decrease in the fluorescence of normal Orai in neuronal somata, in contrast to its increase in distal dendrites, upon exposure to CCCP (Figure 3C), possibly demonstrating the mechanism of Orai1 redistribution caused by SOCE activation [16]. It should also be noted that newly formed filopodia/spines were dynamically visited by Orai1 puncta, in agreement with our earlier observations [17]. On average, the size of STIM1 puncta decreased under CCCP (Figure 3D–F).

To rule out a possible impact of overexpressed phenotypes of the constructs used in this study (morphological markers and versions of Orai1), we compared the level of [Ca^2+^]i in non-transfected cells exposed to CCCP with transfected ones (Appendix A). Without CCCP, transfected cells were spontaneously active and did not reveal a tendency to increase [Ca^2+^]i level during the entire period of recording (Appendix A, ibid). Following CCCP, [Ca^2+^]i levels increased to the same extent and with about the same kinetics as non-transfected neurons (Appendix A). Therefore, we conclude that transfection by itself has no effect on [Ca^2+^]i.

In addition to a sharp increase in [Ca^2+^]i, long-term treatment with CCCP also led to an increase in mitochondrial calcium ([Ca^2+^]m). We detected an initial weak followed by a sharp increase in the fluorescence of the mitochondrial calcium sensor mtRCaMP against the background of a gradual increase in [Ca^2+^]i (Figure 4A). A significant increase in [Ca^2+^]m was recorded at 30 min of treatment with the drug (Figure 4B). The time-lapse recording of new protrusion outgrowth and [Ca^2+^]m rise following CCCP is shown in Appendix A. It is interesting to note that occasionally we observed an “ignition” of mitochondria along the dendrite. An example of such a chain-like increase is shown in Appendix A. Another effect of CCCP on mitochondria was a significant decrease in the length of mitochondrial clusters, detected using the morphological mitochondrial sensor mtDsRed (Figure 4C). After 1 h with CCCP, mitochondrial clusters significantly decreased in length, acquiring a ring/round shape (Figure 4C,D).

We also tested whether apoptotic effects were associated with long-term imaging by keeping control cells and cells treated with 10 µM CCCP at room temperature for 2 and 5 h (Appendix A). The calculation of live cells in control and with CCCP after 5 h showed a significant decrease in the number of healthy cells on coverslips with CCCP (Appendix A). The number of dead cells was significantly higher with CCCP after 2 h and especially 5 h of exposure (ibid). The described visualization method gives a good idea of the advanced stages of cell death, when their structural destruction is visually obvious. Therefore, the effect is best manifested only after five hours of incubation. A more sensitive method for assessing acute cell death, the dead/live analysis [18,19], makes it possible to observe signs of death much earlier. To do this, we loaded cells with Calcein AM, a cell morphology marker, and performed the assay in the presence of propidium iodide (Figure 4E,F). A feature of propidium iodide is that it does not cross the intact cell membrane. However, after perforation of the membrane in diseased cell, the substance enters the nucleus and intercalates between DNA strands. This method made it possible to determine which cells were alive before and died after exposure to CCCP. The percentage of live cells after 3 h of exposure to CCCP decreased from 91% to 44%, while in the control medium, the decrease was only from 87% to 72%. The percentage of dead cells at the start of the experiment in the control media was 13 and 9%, after 3 h, 28% for the control and 56% for the CCCP: control/control n.s., *p* > 0.05; after 3 h: control/CCCP 3 h ***, *p* < 0.001 (Figure 4G,H).

## 3. Discussion

The carbonyl cyanide proton ionophore CCCP has been studied in various cell types. The main target for CCCP is the mitochondria: CCCP is used to study mitochondrial morphology, membrane potential, mitochondrial calcium and ATP production. The CCCP effect on [Ca^2+^]i has been shown in several studies of neurons [20,21,22] and non-neuronal cells [23,24,25,26]; however, no effect of CCCP on neuronal morphology has been demonstrated. Here, we show the effect of sustained CCCP-induced [Ca^2+^]i rise on dendritic morphology. In our recent article, we demonstrated that the application of 10 µM CCCP for 10 min completely abolished global calcium events in cultured hippocampal neurons [14]. However, longer-term treatment with CCCP led to abnormally high [Ca^2+^]i levels, which were accompanied by the appearance of new filopodia and the growth of existing processes. This increase in [Ca^2+^]i is probably due to the influx of extracellular calcium into the cell since the effect does not appear in the nominal absence of calcium in the imaging medium. The CCCP effect of calcium influx was not blocked with APV (selective NMDA receptor antagonist) + TTX (a sodium channels blocker) + DNQX (a competitive antagonist of AMPA receptors) with the help of which excitatory synaptic potentials and backpropagation action potentials, and activity-induced calcium transients were blocked. In addition, the application of nifedipine (L-type VGCC blocker [27]) and ω-conotoxin (N-type VGCC blocker [28] did not block these CCCP effects. There was only a slight [Ca^2+^]i decrease with thapsigargin, a SERCA pump inhibitor against the background of CCCP (Figure 2C). However, high concentrations (50 μM) of 2-APB, a blocker of Orai1,2-mediated Ca^2+^ entry [29], prevented the increase in cytosolic calcium, as well as the growth of spines/filopodia.

We measured the number of new processes with CCCP in categories; filopodia/thin, mushroom and stubby spines. The majority of new processes were filopodia and a few stubby and mushroom spines (Appendix A). However, already existing spines/filopodia may undergo morphological changes with CCCP, as a result of which their assignment to one or another group would change (Figure 1B; Appendix A).

The present results demonstrate that exposure to CCCP for 20–30 min can cause a dramatic effect on the formation of dendritic filopodia via a sustained rise in [Ca^2+^]i. These results do not allow us to speculate at the present time if these processes are to be converted to functional dendritic spines since further treatment with CCCP leads to apoptotic swelling and cell death, as shown in the panels of Figure 2E,F and Figure 4F. In earlier studies, we [13] and others [30] suggested that in normal conditions, filopodia are produced at growth cones as a means to attract the parent neuron to nearby excitatory afferents, and once they attach to these afferents, they convert to functional synapses. Further experiments are needed to explore new grown/overgrown filopodia/spines.

It is noteworthy that under CCCP, the [Ca^2+^]m increased at different rates during the experiment (Figure 4A). In addition, this calcium increase was not observed simultaneously in all mitochondrial clusters but appeared sequentially (Appendix A). Concerning the effect of CCCP on the length of mitochondrial clusters, our results are consistent with previously demonstrated studies [31] that showed that the morphological transition from “filamentous” to shortened, rounded structures depends on [Ca^2+^]i, and the use of a protonophore uncoupler leads to mitochondrial fission and/or sharp shortening/rounding or fragmentation of mitochondria, as well as to the formation of circular mitochondria. CCCP makes the mitochondrial inner membrane permeable to protons, with the result that electron transfer through the electron transport chain is no longer associated with ATP formation due to the loss of the electrochemical gradient [32]. Direct interference of CCCP with mitochondrial functions induces apoptosis through ultrastructural disruption of mitochondria and dissipation of mitochondrial membrane potential, triggering different stress pathways. CCCP-induced depletion of the antioxidant glutathione in mitochondria leads to increased oxygen species production with subsequent cell death. However, for a limited time, cultured cells are able to adapt by developing protective mechanisms against CCCP-induced apoptosis [33]. This study also confirms earlier reports on the involvement of Orai1 and STIM1 in the mechanisms of apoptosis [34] and sheds light on concomitant processes of CCCP-induced apoptosis occurring in neurons that previously went unnoticed. Since oxidative stress and mitochondrial dysfunction are aggravating factors in many neurodegenerative diseases, more investigation is needed to test the role of [Ca^2+^]m in cell death.

The contrast between the early filopodial growth and the later apoptotic process is interesting indeed. After all, other plasticity processes associated with spine formation are not followed by cell death. To examine more closely the transition from spine/filopodia formation to apoptosis, we employed the continuous recording of synaptic currents in cells exposed to CCCP. While the analysis of this study is still in progress, and will hopefully be presented in a follow-up study, suffice it to say that spontaneous miniature synaptic currents increase in frequency over time after initial exposure to the drug, in association with the growth process, and not with the apoptotic process, but a clearer picture of this change awaits further analysis.

Finally, with regard to the formation of new filopodia or spine-like processes, we have shown in previous publications [13,17] that this growth occurs with the participation of the STIM–Orai system, due to which peculiar “hot spots” are created on the surface of dendrites. In essence, they are localized zones of calcium entry into the cytosol through the STIM–Orai channels, which cause the growth of new filopodia. In the cases described above, CCCP seems to act only as a trigger for massive growth, which in itself does not depend on the presence of the drug.

## 4. Materials and Methods

Cultures: Animal handling was conducted in accordance with the guidelines published by the Institutional Animal Care and Use Committee of the Weizmann Institute and with the Israeli National guidelines on animal care. Cultures were prepared as detailed elsewhere [35,36]. Briefly, E17 rat embryos were removed from pregnant decapitated mothers’ wombs under sterile conditions, decapitated, and their brains removed, and the hippocampi were dissected free and placed in a chilled (4 °C), oxygenated Leibovitz L15 medium (Thermo Fisher Scientific Inc., Ness Ziona, Israel) enriched with 0.6% glucose and gentamicin (20 µg/mL, Sigma, St. Louis, MO, USA). Tissue was mechanically dissociated with a fire-polished Pasteur pipette and passed to the plating medium consisting of 5% heat-inactivated horse serum (HS), 5% fetal calf serum (FCS), prepared in MEM-Earl salts (Biological Industries, Beit Haemek, Israel), enriched with 0.6% glucose, gentamicin, and 2 mm glutamax. About 10^5^ cells in 1 mL medium were plated in each well of a 24-well plate onto polylysine-coated 13 mm circular glass coverslips. Cells were left to grow in the incubator at 37 °C, 5% CO_2_.

Plasmids: Neurons were transfected with Orai1-GFP, mutant Orai1 [37] (a gift from Dr. E. Reuveny, the Weizmann Institute), STIM1-mCherry, DsRed or mtRCaMP, eBFP or eGFP (to image cell morphology), using lipofectamine2000 at 1 μL per well with 50 μL per well Opti-MEM (Thermo Fisher Scientific Inc., Ness Ziona, Israel), at 6–7 days in vitro (DIV) and were used for imaging at 10–21 DIV, depending on experiment. The transfection methodology was adopted from standard protocols [13].

Imaging: Fluo-2 AM (2 µM, Thermo Fisher Scientific Inc., Ness Ziona, Israel) or Calcium Orange AM was incubated for 1 h at room temperature to image variations in [Ca^2+^]i. Cultures were then placed in the 3 mL perfusion chamber on the stage of an upright Zeiss 880 confocal microscope using a 40× water immersion objective (1.0 NA) and imaged at a rate of frame/3, 5 or 10 s. No photobleaching was detected under these conditions. Standard recording medium contained (in mM); NaCl 129, KCl 4, MgCl_2_ 1, CaCl_2_ 2, glucose 10, HEPES 10, pH was adjusted to 7.4 with NaOH and osmolality to 320 mOsm with sucrose. All measurements were conducted with identical laser parameters for all groups (e.g., intensity, optical section, duration of exposure and spatial resolution) at room temperature.

Drugs: Preincubation with blockers tetrodotoxin (TTX, 1 μM, Alomone Labs, Jerusalem, Israel) + 2-amino-5-phosphonovaleric acid, (APV) 30 μM) and 6,7-dinitroquinoxaline-2,3-dione (DNQX, 10 μM), thapsigargin 1 μM, nifedipine (1 μM) + ω-conotoxin (50 nM), 2-aminoethoxydiphenyl borate, (2-APB, 50 μM), all from Sigma-Aldrich, were made 10 min before the start of the recordings with CCCP (10 μM). For the wash experiment, the drug was considered to be completely washed out of the chamber after replacing the entire volume of the chamber three times.

Assessing acute cell death, the dead/live assay: Cells are initially loaded in 3 mL standard recording medium with pre-incubation 2 µM calcein-AM in the presence of 2.5 µM propidium iodid (PI) at room temperature and imaged on a stage of a Zeiss LSM 880 confocal microscope using a 20× water immersion objective (1.0 NA). Time-lapse imaging of cultures is performed at room temperature for 180 min at 1 min intervals. Two-channel images [Calcein AM/PI] are acquired, where green (488 nm) and red (545 nm) fluorescent cells represent live and dead cells, respectively. Percentage of live and dead cells is calculated: live cells (cells stained green) or dead/dying cells (cells stained red) multiplied by 100% and divided by the total number of cells (green and red) in the field 200 × 200 micron at *t* = 0. Cells that either lose their color (Calcein-AM leaked out) or gain red staining (PI entering the cell) are not regarded as live (Appendix A). Standard imaging software (Image J 1.52p, NIH, Bethesda, MD, USA) is used to count the number of live cells for each image.

Fluorescence and statistical analysis: High-resolution fluorescent images were analyzed using Image-J (1.52p, NIH, Bethesda, MD, USA) and MATLAB (R2010b, MathWorks, Inc., Natick, MA, USA)-based programs. Statistical comparisons were made with post hoc tests using ANOVA and *t*-tests, using MATLAB (R2010b, MathWorks, Inc., Natick, MA, USA), KaleidaGraph (v. 4.5, Synergy Software Inc, Reading, PA, USA), and OriginPro 2021 (9.8.0.200, Electronic Arts, Inc., San Mateo, CA, USA) software. Statistically significant differences were considered at *p* < 0.05. Dendritic spines/filopodia were identified in the eBFP or DsRed-transfected neurons and analyzed independently of the measurements of calcium.

## Figures and Tables

**Figure 1 ijms-24-12940-f001:**
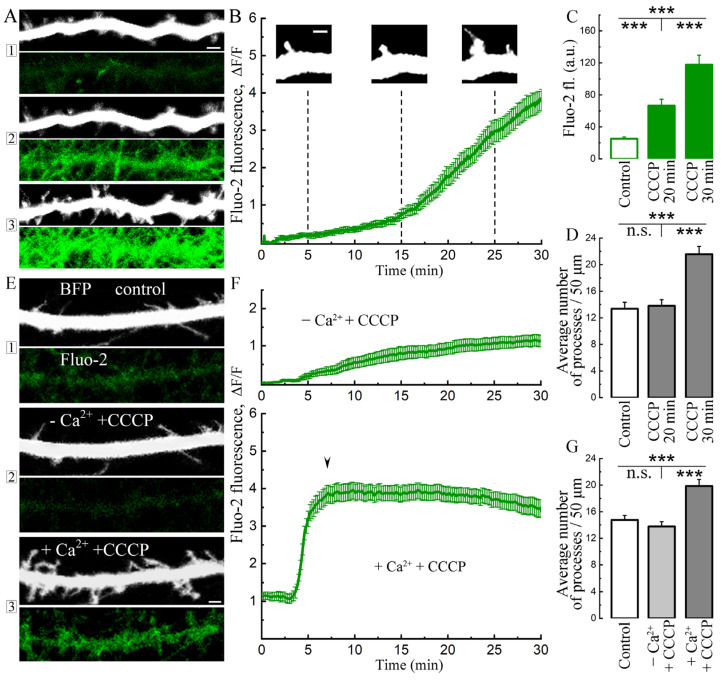
Effect of carbonyl cyanide chlorophenylhydrazone (CCCP) on cytosolic calcium and growth of dendritic processes in hippocampal neurons. (**A**) Dendritic segment from a neuron transfected with eBFP as a morphological marker (up, monochrome) and with Fluo-2 (bottom, green), cytosolic calcium sensor (1—control, 2—20 min and 3—30 min with CCCP 10 μM). Scale 2 μm. (**B**) The panel shows the mean (green trace) from 15 cells (bars indicate SEM). The calcium rise starts after 10–15 min of incubation with CCCP, at the beginning of which rare spontaneous calcium activity may still be observed. By 25 min, the beginnings of new processes appear, which show a sharp growth during the next 5 min of incubation. Scale 1 μm. (**C**) Averaged Fluo-2 fluorescence in control conditions, 20 min and 30 min after treatment with CCCP. Control (normal medium): 25.13 ± 2.18, 20 min CCCP: 66.46 ± 8.04, 30 min CCCP: 117.85 ± 11.77 a.u.; F = 32.13; ***—*p* < 0.001, ANOVA, Bonferroni post hoc, *n* = 15 neurons (somata). (**D**) Averages of dendritic processes detected with eBFP: control: 13.37 ± 0.96, 20 min CCCP: 13.8 ± 0.94, 30 min CCCP: 21.57 ± 1.19; F = 19.92; ***—*p* < 0.001, ANOVA, Bonferroni post hoc, *n* = 15 neurons, a comparable number of dendritic sections, 50 µm long, were analyzed for each cell. For (**B**–**D**), cells from four cell cultures DIV 10–14 were used. (**E**) Low calcium in the imaging solution (2), no effect of CCCP on cytosolic calcium and processes growth was observed (markers as in (**A**)). When calcium returned to the imaging medium, growth of processes was detected (3). Scale 2 μm. (**F**) Only a slight increase in Fluo-2 fluorescence in low-calcium imaging medium was observed over a similar time course of the experiment, as in (**B**) (green curve upper). When calcium returned (2 mM) to the imaging medium (same experiment), after 3–5 min, a sharp, abnormal increase in the cytosolic calcium level was observed (bottom, green curve). The beginning time of spine/filopodia growth marked with arrowhead. (**G**) Quantification of dendritic processes detected per 50 μm dendrite length using eBFP signal in normal medium compared to low-calcium medium and same medium, when calcium was added: control: 14.75 ± 0.68, −Ca^2+^ + CCCP: 13.78 ± 0.71, Ca^2+^ back + CCCP: 19.88 ± 0.98; n.s.—not significant, F = 16.72; ***—*p* < 0.001, ANOVA, Bonferroni post hoc, *n* = 20 neurons from three cell cultures DIV 10–14.

**Figure 2 ijms-24-12940-f002:**
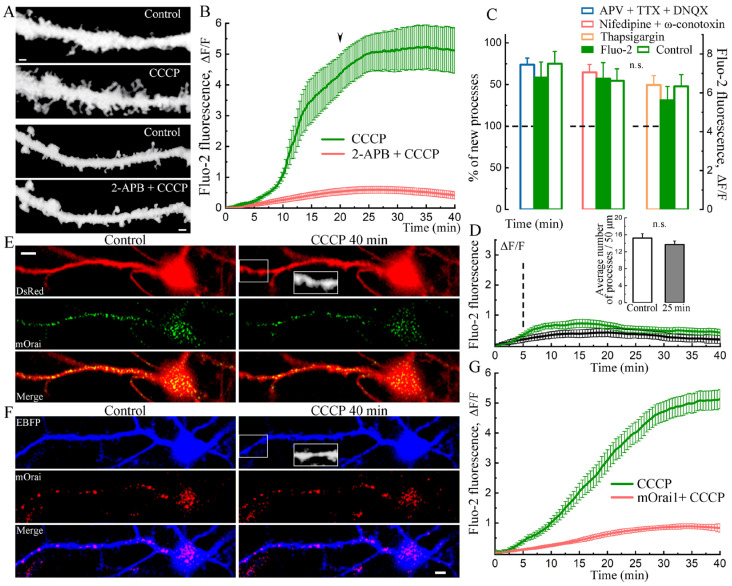
Effect of CCCP through the rise of [Ca^2+^]i on the growth of dendritic processes is dependent on Orai. (**A**) Growth of processes before and after 30 min of incubation with CCCP (eBFP, top two panels) in the presence of 2-APB 50 μM (two lower panels). Scale 2 μm. (**B**) The effect of 2-APB 50 μM (*n* = 16 cells, carrot curve) on [Ca^2+^]i following CCCP. Control-transfected cells (eBFP) with CCCP only: *n* = 15 cells, green curve, start of processes growth marked with arrowhead; three cell cultures, DIV 10–14. (**C**) CCCP-induced rise in cytosolic calcium and dendritic processes is not inhibited by co-incubation with the following blockers: thapsigargin 1 μM, nifedipine 1 μM + ω-conotoxin 50 nM, APV 30 μM + TTX 5 μM + DNQX 10 μM. Left bars of each set are measured in percent of existing/new processes, divided by dotted line. The percentage increases after 40 min of incubation with CCCP with the background of an increase in cytosolic calcium in transfected cells (green bars in group sets, *n* = 18 cells for each) and non-transfected cells (green right bars, *n* = 18 cells for each group). ANOVA, Fisher Post Hoc, three cell cultures. (**D**) Cells were exposed to CCCP for 5 min and then (vertical mark) washed out (16 control-transfected cells, green curve; 24 control non-transfected cells, black curve). After washing, [Ca^2+^]i approached basal levels, and there was no process growth. Bars for control-transfected (eBFP) cells, *n* = 16 cells, three cell cultures DIV 11–15. n.s.—not significant. (**E**) Cell transfected with mutant Orai1 (mOrai1) (green) did not grow dendritic processes in response to CCCP but showed apoptotic swelling after 40 min of treatment (zoomed-in white rectangle). Scale 5 μm. (**F**) Cell transfected with red mOrai1 did not grow spines or filopodia but showed apoptotic swelling after 40 min of CCCP treatment (zoomed-in white rectangle). Scale 5 μm. (**G**) [Ca^2+^]i after CCCP treatment of cells transfected with mOrai1 (red curve, *n* = 16 cells) and control non-transfected cells in sight (green curve, *n* = 36 cells), three cell cultures DIV 10–14.

**Figure 3 ijms-24-12940-f003:**
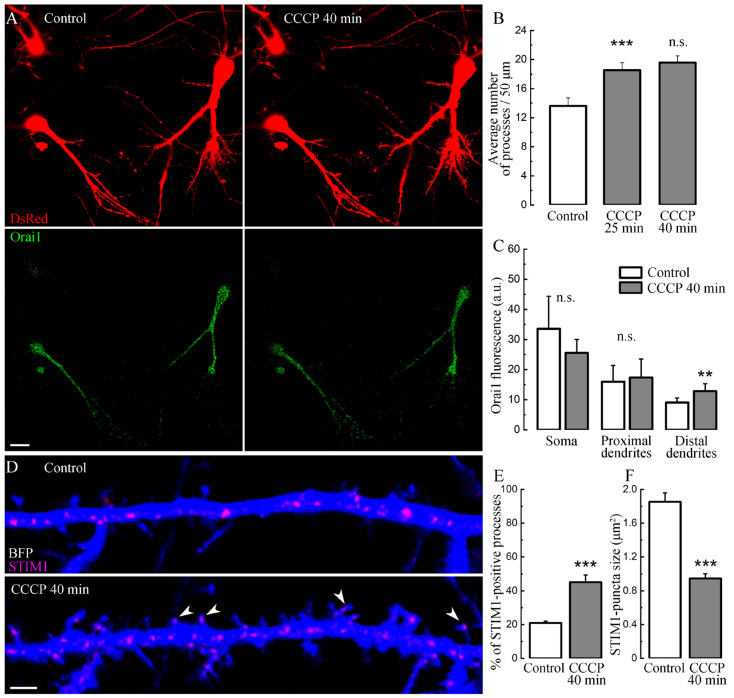
Effect of CCCP on Orai1/STIM1 fluorescence. (**A**) Cells in control medium and after 40 min with CCCP, which are transfected morphological marker DsRed (two upper panels). The same cells were transfected with normal Orai1 (green, two bottom panels, in control medium and after 40 min of CCCP). Scale 10 μm. (**B**) Number of new processes in control and after 25 and 40 min with CCCP, *n* = 15 cells, three cultures, ***—*p* < 0.001, n.s.—not significant, ANOVA, Bonferroni post hoc. (**C**) Orai1 fluorescence level in control and after 40 min of CCCP treatment in soma, proximal and distal dendrites, *n* as in B, **—*p* < 0.01, *t*-tests. (**D**) Dendrite section from a neuron transfected with BFP and STIM1 in control and after 40 min incubation with CCCP. STIM1-puncta translocated into existing and newly formed processes. Scale bar 5 μm. (**E**) Percentage of STIM-1-positive processes in control and after CCCP (20.97 ± 1.06/45.14 ± 4.28 ***—*p* < 0.001, *t*-test). (**F**) The average STIM1-puncta in control and after 40 min of CCCP. For (**E**,**F**) *n* = 18 cells, *t*-test, three cell per culture DIV 18–21.

**Figure 4 ijms-24-12940-f004:**
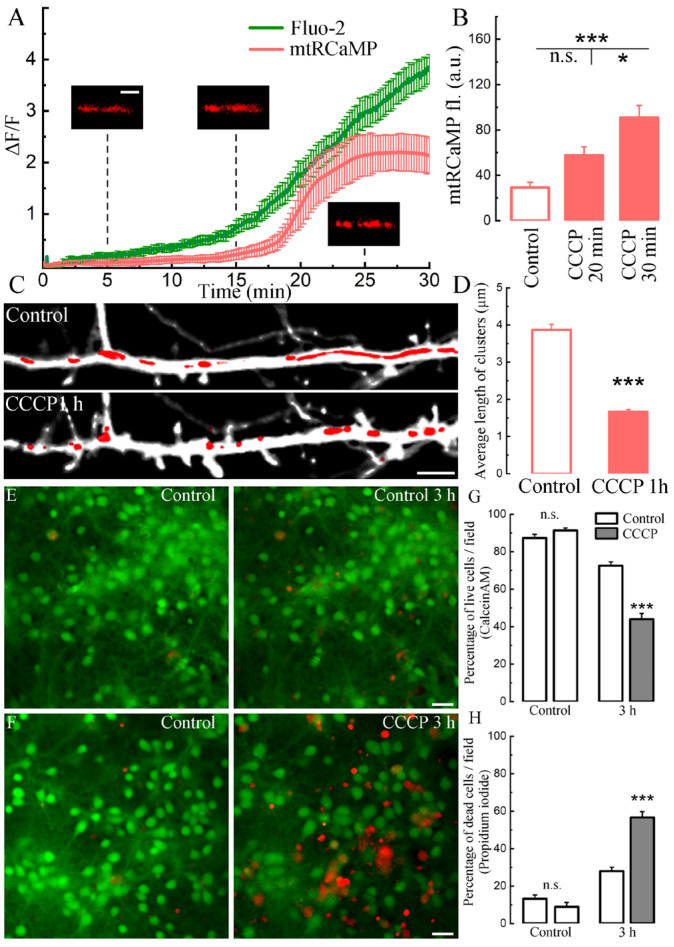
Effect of CCCP on mitochondria and cell survival. (**A**) Mean Fluo-2 (green trace, from Figure 1B, same experiment) conjugated with mtRCaMP fluorescence from 8 cells (bars indicate SEM). Small panels scale 2 μm. (**B**) Averaged mtRCaMP fluorescence in control conditions, 20 min and 30 min after treatment with CCCP. Control (normal medium): 29.15 ± 4.62, 20 min CCCP: 57.67 ± 7.47, 30 min CCCP: 90.88 ± 10.7 a.u.; F = 14.95; *—*p* < 0.05, *** *p* < 0.001, n.s.—not significant, ANOVA, Bonferroni post hoc, *n* = 15, four cell cultures DIV 10–14. (**C**) Dendrite segment in control and under CCCP conditions for 1 h. Morphological sensor eBFP (monochrome), mitochondrial morphological sensor mtDsRed (red). Scale 5 µm. (**D**) Average length of mitochondrial clusters (µm) in control and after 1 h of treatment with CCCP (3.87 ± 0.15/1.67 ± 0.06, ***—*p* < 0.001), *n* = 15 cells, *t*-test, three cell cultures DIV 10–14. (**E**,**F**) Cells loaded with calcein AM (green), a marker of viable cells and propidium iodide, a marker of dead cells (red) in control and after 3 h with CCCP. Scale bar 20 μm. (**G**) Percent of living cells (with calcein-AM) in control medium and with CCCP before and after 3 h of imaging at room temperature (RT). Control/Control: n.s.; Control 3 h/CCCP 3 h: ***. (**H**) Percent of dead cells (with propidium iodide) in control medium and with CCCP before and after 3 h of imaging. Control/Control: n.s.; Control 3 h/CCCP 3 h: ***. For (**G**,**H**), values are mean ± SEM, *n* (control) = 52, *n* (CCCP) = 60 fields, n.s.—not significant, ***—*p* < 0.001, *t*-tests.

## Data Availability

The data presented in this study are available within this paper and the Appendix A.

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
