# Peer review of "Exposure of Cultured Hippocampal Neurons to the Mitochondrial Uncoupler Carbonyl Cyanide Chlorophenylhydrazone Induces a Rapid Growth of Dendritic Processes"

_ijms, 2023, doi:10.3390/ijms241612940_

Round 1
Reviewer 1 Report (Previous Reviewer 2)
Dear authors,
I think the manuscript is stronger than before, with higher sample size coming from multiple independent cultures (as it is commonly found in the literature). Although most of the conclusion have not changed, some of them have actually did (i.e. Figure 2 panel C, Figure 3 panel C, Figure 4 panel H). That was also why I raised the point of the importance of a higher sample size and doing multiple independent experiments/cultures. I hope the authors can appreciate that it was not unjustified and that now the conclusions are better supported.
Best Regards,
Reviewer 2 Report (Previous Reviewer 1)
I am glad to see that the paper has been improved and is now ready to be published.
This manuscript is a resubmission of an earlier submission. The following is a list of the peer review reports and author responses from that submission.
Round 1
Reviewer 1 Report
The paper “Exposure of cultured hippocampal neurons to CCCP induces a rapid growth of dendritic processes· by Kushnireva et al., shows that mitochondrial uncoupler CCCP produces calcium elevations, extensive neurite growth, filopodia and spine production in cultured neurons. The CCCP mitochondrial uncoupler seems to act on Orai1 calcium mobilization.
The paper is well-written and presented. The figures and results are clear, and the potential mechanism is interesting. I have only the following minor concerns:
1. The use of CCCP in the title results is dangerous since it is unknown to a wide audience. I suggest either eliminating the name or explaining what it is.
2. The significance of the results is based on the observations on neurite and filament/spine extension. However, multiple other aspects of neuronal function ay be affected. For example, the group of Spitzer has shown specification of transmitter phenotype and electrical activity. I suggest discussing this possibility in the discussion.
3.- The CCCP activation of mitochondria through influx of calcium may be producing a synthesis of ATP and transport of membrane to the neurites, filopodia and spines. It is well known that calcium accelerates ATP production by acting on several enzymes of the Krebs cycle.
3. Are these observations physiological and therefore significant to neurons? The apoptosis described by the authors may mean a toxic effect that could be discussed deeply.
Author Response
- The use of CCCP in the title results is dangerous since it is unknown to a wide audience. I suggest either eliminating the name or explaining what it is.
- Thank you, now we have changed the title to add “…the mitochondrial uncoupler CCCP…”.
- The significance of the results is based on the observations on neurite and filament/spine extension. However, multiple other aspects of neuronal function ay be affected. For example, the group of Spitzer has shown specification of transmitter phenotype and electrical activity. I suggest discussing this possibility in the discussion.
- Thank you, we have now added discussion on electrophysiological consequence of exposure to CCCP. The results of these experiments will be included in a follow-up study, which we are now summarizing for submission
3.- The CCCP activation of mitochondria through influx of calcium may be producing a synthesis of ATP and transport of membrane to the neurites, filopodia and spines. It is well known that calcium accelerates ATP production by acting on several enzymes of the Krebs cycle.
- Indeed, we discussed these possibilities in the added discussion section.
- Are these observations physiological and therefore significant to neurons? The apoptosis described by the authors may mean a toxic effect that could be discussed deeply.
- Thank you for the suggestion. We tested the time course of the effects of CCCP to find that a short exposure to the drug did not cause persistent damage to the cell, and only long term exposure, far after the morphological effect, did the cells undergo apoptosis (Fig 2).
Reviewer 2 Report
In the manuscript titled “Exposure of cultured hippocampal neurons to CCCP induces a rapid growth of dendritic processes” Kushnireva and colleagues investigate the role of Carbonyl Cyanide Chlorophenylhydrazone (CCCP), a mitochondrial uncoupler, in dendritic growth using neuronal cultures. Even though some aspects of the study are interesting, including the use of live imaging, the main conclusions of the manuscript are not well supported by the data, especially due to the lack of controls and the sample size. Please see below all the details regarding this in the following major and minor concerns:
Major concerns:
1) According to materials and methods, cultures were used for imaging at 10-21 DIV. This is an extremely broad range, and the maturation of neurons is very different during that period. The authors should clarify which DIV is analyzed in each figure and be consistent of this DIV in experiment of the same type.
2) The number of independent cultures analyzed is missing. How many independent experiments were performed? Do all experimental measurements come from one culture? Analyzing several neurons from 3 independent cultures is the standard to reach the appropriate biological variability and statistical power. Experimental conclusions cannot be supported by a sample size of 3-8 neurons coming from the same culture. Also, the authors use dendritic segments as the n in one of the experiments, instead of neurons, which would be the appropriate sample.
3) It is not clear what the authors measure as dendritic processes. Are the authors measuring filopodia and/or dendritic spines? It looks like they are quantifying all processes together, but they are part of distinct steps of synapse formation and require distinct Calcium requirements and timing. Normally filopodia-like protrusions are dynamic and are more than 10um long. Dendritic spines are less than 10um long and the shape of the spine is linked to the maturity of the process.
4) The authors discuss several important controls and results by mentioning them in the text but not showing graphs and/or described as “data not shown”. They should include all this data in the figures.
5) In Figure 2 and 3, cells not transfected with and without CCCP treatment should be included as controls and compared to the other experimental treatments, to rule out overexpression phenotype of the constructs used in the study (mutant and wild type versions of Orai1).
6) In Figure 4 the authors should analyze neuronal death by using specific staining for cell death such as CC3, TUNEL or necrosis markers; also it would be helpful checking the nuclear morphology. Moreover, it is not surprising that leaving the neurons at room temperature with 10 μM CCCP will cause massive neuronal death. Can the authors make this experiment at 37C with 5% CO2 in the incubator?
Minor comments:
1) There is a mistake in materials and methods, the authors should replace the term “pregnant rats” for “rat embryos” in the sentence “Briefly, E17 pregnant rats were decapitated, their brain removed”.
2) Explaining briefly all the drugs (i.e., mode of action) used in the study and the rationale of some experiments, would improve the flow of the manuscript.
3) Material and methods section is incomplete; some details of the study are missing, including the details of the drugs used.
4) In Figure 2 authors mention that none of the used drugs inhibit CCCP effect; however, in the same figure legend, they clarify that significant reduction was only seen using Thapsigargin (also shown in the graph of Figure 2C). How do authors explain this discrepancy?
5) The data in Figure 3 is very interesting. However, it is not clear how the authors differentiate between proximal and distal dendrites.
6) In Figure 4 it is not clear if the clusters result from mitochondria fusion or lack of mitochondria fission.
No comments on the quality of English Language.
Author Response
1) According to materials and methods, cultures were used for imaging at 10-21 DIV. This is an extremely broad range, and the maturation of neurons is very different during that period. The authors should clarify which DIV is analyzed in each figure and be consistent of this DIV in experiment of the same type.
- We would like to clarify that the vast majority of experiments were carried out in DIV 10-14 cultures. An exception is the STIM1 experiment, where we use a 18-21 DIV cultures to enhance the contrast of STIM1 entry into the spines (according to our previous experiments: Kushnireva et al., 2021). We have now labeled the culture age for all experiments individually.
2) The number of independent cultures analyzed is missing. How many independent experiments were performed? Do all experimental measurements come from one culture? Analyzing several neurons from 3 independent cultures is the standard to reach the appropriate biological variability and statistical power. Experimental conclusions cannot be supported by a sample size of 3-8 neurons coming from the same culture. Also, the authors use dendritic segments as the n in one of the experiments, instead of neurons, which would be the appropriate sample.
- We apologize for missing details about the number of cell cultures. Indeed, we followed repeatability of the experiment in three cell cultures (except for some additional confirmatory sets of experiments). To obtain data from one transfected cell, in most cases a whole cover glass is used, and so our samples are not very large. However, our findings are confirmed in independent experiments in several dissections. We now caption the number of cell cultures, added experiments, corrected inaccuracies and show only the number of neurons as n. In total, 82 independent experiments were conducted in this study.
3) It is not clear what the authors measure as dendritic processes. Are the authors measuring filopodia and/or dendritic spines? It looks like they are quantifying all processes together, but they are part of distinct steps of synapse formation and require distinct Calcium requirements and timing. Normally filopodia-like protrusions are dynamic and are more than 10um long. Dendritic spines are less than 10um long and the shape of the spine is linked to the maturity of the process.
- Indeed, we use the term "processes" for both spines and filopodia. In the revised manuscript, we also did a quantitative analysis of processes: filopodia, mushroom and stubby spines, and added the data in the Supplementary figure, D. The criticism of this analysis may be in the fact that during the CCCP exposure, not only new spines / filopodia appear, but the morphology of existing ones also changes. We can no longer judge unequivocally their belonging to a certain category (see Supplementary figure, C).
4) The authors discuss several important controls and results by mentioning them in the text but not showing graphs and/or described as “data not shown”. They should include all this data in the figures.
- Thank you for your attention to detail. We have now enriched the text with the relevant data.
5) In Figure 2 and 3, cells not transfected with and without CCCP treatment should be included as controls and compared to the other experimental treatments, to rule out overexpression phenotype of the constructs used in the study (mutant and wild type versions of Orai1).
- At the request of the reviewer, we included data from not-transfected cells with and without CCCP as a control for Figure 2, B (Supplementary figure, A); data from not-transfected cells for experiments in Figure 2, C and D; we also clarify that for Figure 2, G, non-transfected cells were used as controls. We did not use Fluo-2 calcium indicator in experiments with Orai1 on Figure 3 due to the fact that the green channel was already occupied. Now, at the request of the reviewer, we added an experimental session with a red cytosolic calcium indicator for cells loaded with morphological marker EBFP and Orai1 and control cells in the field (Supplementary figure, B). Based on the data, we conclude that transfected cells differ insignificantly from non-transfected controls in СССP effects.
6) In Figure 4 the authors should analyze neuronal death by using specific staining for cell death such as CC3, TUNEL or necrosis markers; also it would be helpful checking the nuclear morphology. Moreover, it is not surprising that leaving the neurons at room temperature with 10 μM CCCP will cause massive neuronal death. Can the authors make this experiment at 37C with 5% CO2 in the incubator?
- We have now added a standard method for assessing acute cell death, the dead/live assay (Slepian et al., 1996; Sadeh et al., 2016). In this assay, we load cells with Calcein-AM, which, penetrates live neurons, and converted from a non-fluorescent compound into a highly fluorescent green fluorophore. Subsequently, we expose the neurons to CCCP, in the presence of propidium iodide (PI) which penetrates dead cells, and count red/green fluorescent cells. This method allows us to examine which cells were alive before, and died after exposure to CCCP.
- We would also like to draw attention to the fact that leaving control neurons at room temperature does not cause such massive neuronal death as in СССP (Figure 4, G). Finally, we are using room temperature for recording/imaging neurons, and they are alive for several hours, and their pH is buffered with HEPES. We use this protocol for decades.
Minor comments:
1) There is a mistake in materials and methods, the authors should replace the term “pregnant rats” for
“rat embryos” in the sentence “Briefly, E17 pregnant rats were decapitated, their brain removed”.
- We apologize for this misprint. Now we have corrected and expanded the description of the methods.
2) Explaining briefly all the drugs (i.e., mode of action) used in the study and the rationale of some
experiments, would improve the flow of the manuscript.
- Thank you, we are now adding a briefly summary of all used drugs.
3) Material and methods section is incomplete; some details of the study are missing, including the details
of the drugs used.
- Thank you, now we added details in the ‘methods’ section.
4) In Figure 2 authors mention that none of the used drugs inhibit CCCP effect; however, in the same
figure legend, they clarify that significant reduction was only seen using Thapsigargin (also shown in the
graph of Figure 2C). How do authors explain this discrepancy?
- We have now corrected this misunderstanding in the text. It was meant that the level of calcium with thapsigargin is reduced after the use of СССP in comparison with nifedipine + conotoxin and APV + TTX + DNQX, but this decrease does not cancel the effect of the growth of processes, in contrast to 2-APB, when using which the calcium level after treatment with CCCP does not rise above the control level without CCCP (Supplementary figure, A), and the effect of process growth does not appear at all.
5) The data in Figure 3 is very interesting. However, it is not clear how the authors differentiate between
proximal and distal dendrites.
- We used 2 criteria, depending on the cell morphology: first-order dendrites or dendrites located at a distance of no more than 50 µm from the soma were considered proximal. Second-order dendrites or dendrites located at a distance of more than 50 µm from the soma were considered distal.
6) In Figure 4 it is not clear if the clusters result from mitochondria fusion or lack of mitochondria fission.
- According to the study to which we refer (Brustovetsky et al., 2009), morphology of mitochondria under mitochondrial uncoupler changes due to fragmentation, circulating and shortening. In our experiment, we measure length of visualized mitochondria / mitochondrial clusters, which, on average, significantly decreases under the influence of СССP (Fig. 4, G). We see very similar results: mitochondria fission, no tendency to fusion, rounding / swelling lead to decreasing in length.
Round 2
Reviewer 2 Report
Dear authors,
The manuscript has substantially improved and you have addressed most of my concerns, except the sample size (n), which I think it is still too low to reach substantial support for the conclusions draw in this manuscript. I still think that analyzing more cultures (at least 3) and neurons (more than 15-20 in total) would strengthen the result and conclusions, but if it would take much time for the authors to do that and they prefer leaving it as it is, and the Editor is ok with this, then I will lead readers to judge by themselves how strong they think the conclusions are with the current sample size.
Best Regards,